# Discriminative Candidate Generation
# for Medical Concept Linking

**Elliot Schumacher**                                    ESCHUMAC@CS.JHU.EDU

**Mark Dredze**                                          MDREDZE@CS.JHU.EDU

JOHNS HOPKINS UNIVERSITY

## Abstract

Linking mentions of medical concepts in a clinical note to a concept in an ontology enables a variety of tasks that rely on understanding the content of a medical record, such as identifying patient populations and decision support. Medical concept linking can be formulated as a two-step task; 1) candidate generator, which selects likely candidates from the ontology for the given mention, and 2) a ranker, which orders the candidates based on a set of features to find the best one. In this paper, we propose a candidate generation system based on the DiscK framework [Chen and Van Durme, 2017]. Our system produces a candidate list with both high coverage and a ranking that is a useful starting point for the second step of the linking process. we integrate our candidate selection process into a current linking system, DNorm [Leaman et al., 2013]. The resulting system achieves similar accuracy paired with with a gain in efficiency due to a large reduction in the number of potential candidates considered.

## 1. Introduction

Medical concept linking (aliases: mention normalization, medical concept parsing, biomedical entity linking) associates a medical concept in an ontology with a textual reference in the medical record. This a key task in making free text in electronic medical records accessible for a range of patient care and research goals. A large amount of patient information is encoded in the free text notes associated with a patient, and concept linking allows the concepts mentioned in the note to be formally identified. This processing step allows downstream applications to access a wider variety of information encoded in the notes. Concept linking improves over string matching by identifying concepts (e.g. myocardial infarction) within text that aren't restricted to lexically similar phrases, by identifying phrases that may be lexically distinct but refer to the same concept (e.g. heart attack).

This task can be broken down into a two-step system; a candidate generation step, which produces a list of possible concepts from the ontology, and a ranker, which identifies the most likely candidate concept, often based on machine learning and extracted features. This two-step approach has the benefit of pruning unlikely concept candidates prior to the final linking stage, a necessary step when dealing with ontologies with millions of concepts. This is in contrast to a single-step approach, which chooses a link from the entire ontology, and may require the computation of more fine-grained features over a larger set of concepts. The two-stage approach allows for a simpler feature set to be used in the first step, and a more fine-grained feature set to be used in the final step.

Previous work in concept linking has largely focused on developing rankers (or classifiers), assuming a method that produces a list of concepts which contain the correct answer. Candidate generation must fast, so it often relies on basic lexical matching algorithms that produce a large list of candidates but do not incorporate features or machine learning. Such as system could consist of simple n-gram matching or other lexical similarity features comparing the mention string and the

concept name. Others, such as in the first version of MetaMap [Aronson, 2001], generate possible variations of the mention drawn from the structured information within the ontology, and score each on the type of variation present. While more sophisticated than standard lexical matching, it requires all likely variations to be present in the ontology or annotated dictionaries. Similarly, Aggarwal and Barker [2015] has included a candidate over-generation phase, where possible variants are proposed and then re-ranked by inverse document frequency (IDF).

We propose a candidate generation system that produces a candidate list that has both high coverage, and a ranking that is a useful starting point for a final classifier. We adapt DiscK [Chen and Van Durme, 2017], a framework that allows for feature template-level weighting, and efficient retrieval by feature projection. Using a feature-based system provides flexibility in selecting the criteria for candidate concepts. We consider several different feature templates useful for medical concept linking, and learn a retrieval function. We develop our system for linking disorder mentions, and evaluate using information retrieval metrics that measure the quality of the ranked candidate list. We find that our approach improves over several standard lexical matching baselines. Finally, we integrate our candidate generation system into an existing concept linking system [Leaman et al., 2013]. Although restricting candidates to those generated by DiscK causes a small reduction in coverage and mean reciprocal rank, large gains in efficiency are made due to the reduced number of concepts considered in the final linking stage.

## 2. Triage for Medical Concept Linking

Concept Linking matches previously-identified spans of text (e.g. in clinical notes) to a structured knowledge base of concepts, also known as an ontology (e.g. SNOMED-CT). For example, in the sentence "The child had *juvenile diabetes*", the text mention *juvenile diabetes* references the concept *Diabetes Mellitus, Insulin-Dependent* in the ontology. For each concept in the ontology, there is a Concept Unique Identifier (CUI) (C0011854), one or more names, and other relational information. Concept linking attempts to identity the correct CUI, despite similar candidates (e.g. the concept *Juvenile onset diabetes mellitus, in pregnancy*) and mention variation (*juvenile diabetes* vs. *Diabetes Mellitus, Insulin-Dependent*). If no relevant concept is identified for a span of text, it is annotated as *CUI-less*. Named entity recognition (NER) is a related task that is often combined with Concept Linking in order to first identify potential candidate mention spans within documents. Concept linking has been explored in medical and biomedical domains [Aronson, 2001, Savova et al., 2010, D'Souza and Ng, 2015, Rajani et al., 2017, Zheng et al., 2015], and for other non-medical tasks [Cheng and Roth, 2013, Liu et al., 2013].

The amount of information provided in the ontology is directly related to the difficulty of linking concepts to text mentions. If a only a single concept name is provided, the concept linking system may not be able to match spans that contain variations of the standard concept name. Additional synonyms, including ones that are more commonly used in clinical text, allows for the use of standard string matching. For our ontology, we restricted synonyms to only include preferred entries, so only 22,769 out of 125,362 concepts have at least one synonym included. Most previously developed medical concept linking systems have focused on features that capture the similarity between the concept name and mention text, and have only used the definition text (or other sources of information) as auxiliary information.

While some links can be identified by direct or partial word matching with the concept name in the ontology, many require additional lexical transformations. These include transformations to

account for morphology (e.g. mention *cystic*, concept *cyst*), abbreviations (e.g. *CAP*, *community acquired pneumonia*), and spelling differences (e.g. *pneumoniae*, *pneumonia*). Additionally, many mentions require non-lexical transformations to correctly identify links (e.g. *gastropathy*, *stomach disease*). Many ontologies were constructed using formal language and often do not account for the range of language used in less formal applications like clinical notes.

Many concept linking systems are formulated as a two step problem. First, a candidate generation (or triage) stage selects a subset of the concepts in the ontology that may be good candidates for links. This is often done based on name similarity between the mention string and the concept name. The second step consists of a final linking step that determines which of the previously generated candidates is the most likely concept to be referenced in the mention text. A two step process is often necessary when a system employs machine learning based on features to select the best candidate. It is infeasible to extract features and score every concept in the ontology, so a first step of triage is necessary to reduce the number of possible candidates. Triage is feasible since many likely candidates can be identified by simple attributes, such as string overlap, which are also computationally efficient. Reducing the pool of candidates in triage then allows for more fine-grained attributes to be considered in the final linking stage.

For a mention $m$ containing a concept reference, the candidate generation process for selecting $N$ candidate concepts can be defined as follows. Given a set of concepts $c$ from an ontology $C = (c_0, ..., c_Q)$ containing $Q$ candidates, we can select $N$ candidate concepts from the ontology by scoring each candidate with a candidate likelihood function $f$. Therefore, for each candidate $i$ in $C$, a score can be calculated as

$$\text{score}_i = f(m, c_i) \tag{1}$$

The set of candidates $C$ is then sorted by the candidate score, with the top $N$ candidates from the sorted candidate list selected as the final candidate list.

The scoring function $f$ can be formulated using several different methodologies. Many systems use non-feature based approaches, only considering candidates that match a single criteria. These include string matching algorithms such as the Levenshtein or Jaro-Winkler distance. While this approach benefits from simplicity, it excludes candidates with low string similarity. In contrast, a feature-based approach can consider multiple attributes, such as string similarity, but can be designed to include correct candidates that may not be identified by a simple method by using a variety of features. In general, we would prefer to use a more flexible feature based approach to triage, but are limited in that we cannot efficiently compute features and scores for a large set of candidates.

## 3. Discriminative Information Retrieval for Knowledge Discovery

Chen and Van Durme [2017] introduced the framework *DiscK*, which formulates candidate generation as a feature-based classification retrieval problem. Using a simple feature set, it learns a weighted similarity score for a query and each concept, creating a ranked list of concepts for each query. Given a query $q$ and a candidate set $D = \{p_1, ..., p_N\}$, the system scores the pair by a specified feature function $F(q, p)$ and retrieves the top-k candidates:

$$\underset{p \in D}{\text{argmax}}\{F(q, p)\} \tag{2}$$

This normally requires scores to be calculated between every query and candidate, which is not efficient for larger sets. However, DiscK proposes a feature set formulation that allows for feature

projection – for a given query, the expected feature values for the relevant candidate can be calculated. This allows for efficient retrieval by indexing, and therefore pairwise scoring is not required.

To allow for this efficient retrieval, they restrict features to two feature types[1]. The Cartesian Product, one of the feature types, for a query and a candidate is defined as

$$f_Q(q) \otimes f_P(p) = \{((k_i, k_j) = (v_i, v_j), w_i)\} \tag{3}$$

for a query $F_Q(q) = \{(k_i = v_i, w_i)\}$ and for a candidate $f_P(p) = \{(k_j = v_j, 1)\}$. The variables $k_i$ and $k_j$ refer to specific features type instances and $v_i$ and $v_j$ refer to feature values, and $w_i$ refers to the weight of that feature instance. For example, if the feature type is the bigram word count of the query string *broken leg*, $k_i$ would be the feature instance (e.g. *broken_leg*) and $v_i$ is the feature value (e.g. *1*). The projection of the Cartesian Product is defined as

$$t_\theta^\otimes(f) = \{(k' = v', w\theta_{(k,k')=(v,v')}|k = v, w) \in f\} \tag{4}$$

for all $k', v'$ such that $\theta_{(k,k')=(v,v')} \neq 0$. With this definition, they show that with model parameters $\theta$,

$$t_\theta^\otimes(f) \cdot g = \theta \cdot (f \otimes g) \tag{5}$$

meaning that the projected features of the candidate multiplied by the features of the candidate is equivalent to the weighted pairwise score of the query and candidate. The term $w$ is specified for each feature – this can be any real number in $[0, 1]$, representing a Boolean or a normalized count, for example. The feature parameters $\theta$ are selected to optimize the retrieval equation (noted as Equation 2) on the training data. These parameters $\theta$ are trained using a negative sampling procedure, with the goal of learning a set of weights that will correctly predict mention-concept pairs. For each training mention, it is paired with the correct ontology entry and 50 incorrect ontology entries. The resulting weights are used to project which ontology entry is most suited to the mention feature set. This is trained using a log-linear model. This formulation is computationally efficient because it only involves a sparse feature set, which allows for efficient retrieval. However, it also restricts the types of features.

DiscK allows us to efficiently retrieve candidates over a large ontology in sublinear time, and select a small subset containing likely links. A final linker can then use this subset to make linking decisions. As a ranker only needs to consider a subset of the entire ontology (e.g. 1% of candidates), computationally-intensive features can be used at a smaller total computation cost. In larger sets of clinical notes, this will reduce the total computational cost, making the entire concept linking pipeline more efficient.

## 4. DiscK for Clinical Concept Linking

Using the framework discussed in the previous section, we developed a version of DiscK suited for clinical concept linking. The feature set developed for this task consists of feature templates that capture the relatedness of the mention text and properties of a UMLS concept. While features from additional mention properties, such as the surrounding sentence, were tested, none provided an improvement over features built from the mention span text alone. The features tested included a bag of word template using the entire sentence and a range of ngram sizes for words and char-grams.

---

1. We do not consider one, the Join type, for features in our current system.

We imagine this is the case since the wider sentence context may often not be lexically similar to the concept name or definition.

We used several lexical features to adapt DiscK to concept linking candidate generation. These include the following feature templates.

- A full-string match between the mention and any of the concept names, which receive a feature value of 1 if they are identical.

- A bag of words feature template that matches overlapping individual words between the mention text and the concept name, where each overlapping word is individually weighted by its inverse document frequency[2].

- A bag of words feature template that matches overlapping individual words between the mention text and the concept definition (if present), where each overlapping word is individually weighted by its inverse document frequency. The inverse document frequency weights were calculated on a separate non-medical corpus.

- For some models, a bag of character-grams of length 6 are included (see below) – a range of lengths were tested, but this size resulted in the largest coverage increase on the development set. The resulting character-grams are also weighted by inverse document frequency [3].

- An abbreviation dictionary built from the Wikipedia list of disease abbreviations [4], and matched to the mention text.

- For some models, an expanded abbreviation algorithm was included, which simply combines the first character in each word in the concept name to create an acronym.

- A lemmatized bag of word feature template, using the Stanford Toolkit [Manning et al., 2014], in order to capture any overlapping words that would be excluded due to differences in morphology.

While we found that many mentions could be matched to concepts by lexical features, a significant portion required non-lexical features (e.g. mention *joint pains*, concept *Arthralgia NOS*) Therefore, we added the mention text of any linked concept in the training data to the set of concept names. We then added these mentions to the concept name's bag of word feature template. This addition helped capture some non-lexical matches, but this only assists in non-lexical matches found in the training set.

## 5. Data

We evaluate our system on the concept linking dataset released for ShARe/CLEF eHealth Evaluation Lab 2013 Task 1b [Pradhan et al., 2013]. This dataset consists of concept span annotations built on a subset of MIMIC 2.5 clinical notes [Saeed et al., 2011]. The publicly available training set consists of 200 clinical notes, which we split into a training set consisting of 100 notes (1964 included

---

2. calculated in a separate corpus
3. Character-gram IDF statistics were calculated on the MIMIC corpus.
4. https://en.wikipedia.org/wiki/List_of_abbreviations_for_diseases_and_disorders

mentions), and development and testing sets consisting of 50 notes each (957 and 1076 included mentions, respectively)[5]. We do not report on the task-designated test set as it was unavailable.

Each disorder mention in the clinical note is annotated with concept information. This information either includes the relevant concept unique identifier (CUI), or annotations noting cases where the correct concept could not be identified – primarily with the CUI-less annotation. The annotations guidelines state that the concept candidates should be limited to the SNOMED-CT portion of the Disorder Semantic Group in the Unified Medical Language System version 2011AA [Campbell et al., 1998], and lists the semantic types included in the Disorder Semantic Group. However, we found several annotations linked to concepts not included in that list, including the *Finding*, *Body Substance*, and *Mental Process* semantic types, and therefore we expanded our ontology to include those concepts. Finally, we include all preferred entries, with the default settings of UMLS 2011AA, in the SNOMED-CT Disorder Semantic group (accounting for 116,436 unique concepts), but also include the first non-preferred entries that do not have a preferred entry (accounting for 8,926 unique concepts). We exclude any concept mentions that are not annotated with a SNOMED-CT Disorder concept, including non-concept annotations.

## 6. Evaluation and Results

While we are using (almost[6]) the same dataset as the Share/CLEF 2013 task [Pradhan et al., 2013], we are considering a different task. The systems in the shared task are end-to-end concept linking systems [Pradhan et al., 2013, Savova et al., 2010, Aggarwal and Barker, 2015, D'Souza and Ng, 2015], whereas we consider a candidate generation system. The concept linking systems that were evaluated in the shared task may have included a candidate generation stage, but evaluations of these stages are not provided and we were not able to locate the code of such a system.

Instead, we compare our candidate generation approach to several representative baselines.

- **Exact Match** – Selects concepts that are an exact string match to the mention text.

- **Partial match** – Scores concepts by the number overlapping words that occur between the concept name and the mention text.

- **Char 4-gram** – Scores concepts by the number of character 4-grams that overlap between the concept name and the mention text.

- **BM25** – Scores overlap between the concept name and the mention text using BM25 [Robertson et al., 1995], a common information retrieval method.

- **DiscK Binary** – Uses the same feature set as the weighted models (noted as **DiscK Weighted**), but use binary weights instead of those learned in training. This evaluates the effectiveness of training a model based on these features compared to using un-weighted features.

- **DiscK Combined** – Combines the ranking of the best performing model with respect to mean reciprocal rank and at lower coverage levels (DiscK-1, R=0.4) with the best performing model at higher coverage levels (Char 4-gram). Specifically, we normalized the score of each to be between 0 and 1. For DiscK-1, we performed minmax normalization on each individual

---

5. These numbers do not include excluded mentions. The following paragraph provides further details.
6. We do not have access to the Test set, giving us less overall data and different evaluation sets.

candidate list, and for Char 4-gram, we divided the number of overlapping character-grams by the number present in the mention. For each candidate, we selected the max score between the two models. If only one model assigned a score to a candidate, that score was used.

Two DiscK models are reported – one that excludes character-grams and only uses a dictionary to look up for abbreviation expansion (DiscK-1), and one that includes 6 character-gram features and an expanded abbreviation algorithm (DiscK-2). All systems use synonym augmentation – for each mention (or text span) in the training data that is annotated with a link to a concept in the UMLS, the mention text is added as a synonym to the set of concept names already present in UMLS. This step, which is common in clinical concept linking, allows for additional synonyms to be identified, including those that are likely to only occur within clinical text. However, is limited by the size and diversity of the training data. All systems were trained on both train and development sets for the final tests.

For our implementations and baselines, we report both coverage, the percentage of instances that the relevant concept was generated, and mean reciprocal rank, to measure the effectiveness of the ranking. When calculating the mean reciprocal rank, if any concepts are tied, they are randomly ordered and assigned the corresponding rank. Several regularization parameters were tried on the DiscK model - regularization controls both the weights of the model and the feature selection of the model. The least regularized DiscK-2 model (R = 0.4) uses six feature templates, while the most regularized model (R = 0.25) contains five feature templates. While the weights resulting from regularization are not relevant to the Binary models, the feature template set from the non-Binary version is used in the Binary version.

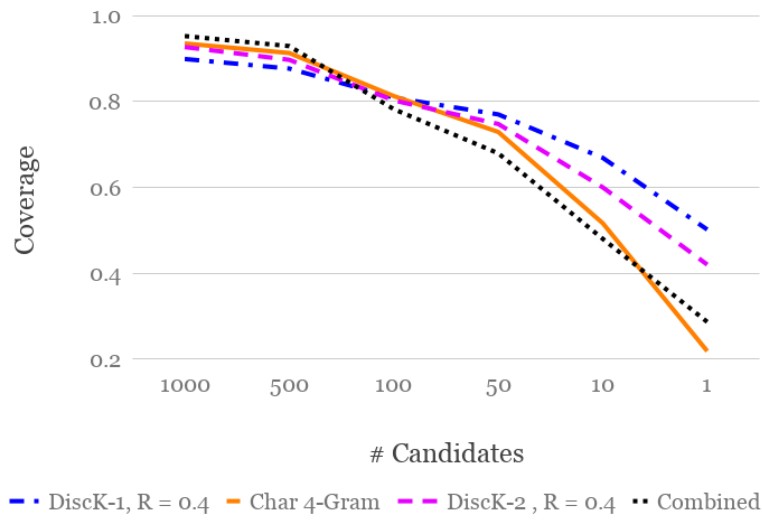

Figure 1: Coverage

As shown in Table 1, although DiscK-2 is competitive, the 4 character-gram baseline performs the best at $K = 100$, and the Combined model performs best at $K = 1000$. However, at smaller candidate sizes (e.g. 10 and 1), DiscK-1 provides the best coverage, representing a 13.9% improvement over 4 character-gram baseline. This change is illustrated in comparing the coverage to the

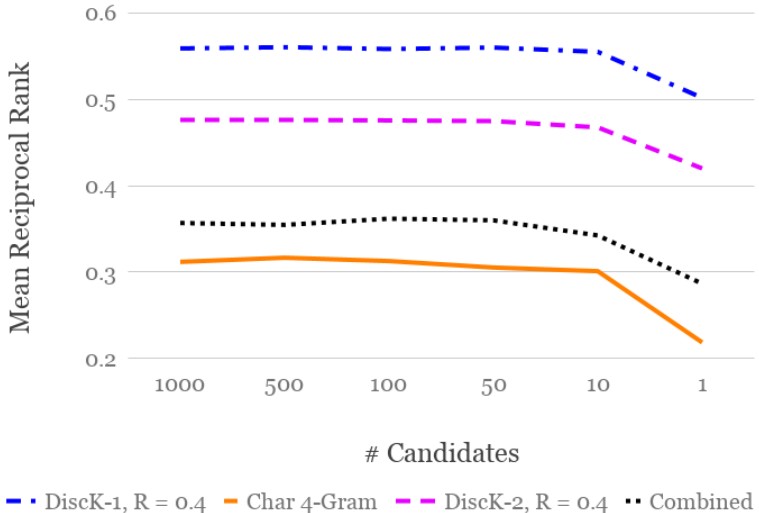

Figure 2: Mean Reciprocal Rank

candidate size in Figure 1. For mean reciprocal rank, we find that the DiscK-1 model provides the best mean reciprocal ranking in all settings. Unlike with coverage, this improvement is clear at each candidate list size, with the best DiscK model at each size providing at least a 0.09 improvement in mean reciprocal rank compared to the binary models, and at least a 0.18 improvement over the non-DiscK baselines. This is illustrated in Figure 2, which compares mean reciprocal rank at different candidate sizes.

We ran feature ablation tests for the DiscK-2 model, with R = 0.25 and N = 1000, shown in Table 3. The most important feature templates are the Partial, Lemma, and 6 character-gram group – the coverage is reduced by half, and the mean reciprocal rank is also reduced. The ontology augmentation step (which is used in the partial matching feature template) is also an important component of the system, as its omission results in a 8.6% drop in coverage. While all feature templates contribute to increased coverage, the 6 character-gram omission results in increased MRR.

To determine whether using learned weights produces rankings that are distinct from those produced by binary models, we used the Wilcoxon signed-rank test to compare each DiscK model with one using binary weights. We find that with the exception of DiscK-2 with R = 0.25, the p-value for the test is less than 0.01, allowing us to reject the hypothesis that the weighted versions produce the same ranking as non-weighted versions.

## 7. Discussion

While several baselines do an equivalent or better job on coverage (including the correct concept in the candidate list), DiscK consistently does a better job of assigning a higher rank to the right link (higher MRR). DiscK achieves at least a 0.09 improvement over binary DiscK in MRR, and at least a 0.18 improvement over all other baselines. Excluding the combined model, the best performing coverage at N = 1000, 4 char-gram, does exceed the best DiscK model by 0.7. However, the best DiscK model for mean reciprocal rank exceeds the 4 char-gram model by 0.18. While many baseline

| | List Size | | $K = 1000$ | | $K = 100$ | | $K = 10$ | | $K = 1$ | |
|---|---|---|---|---|---|---|---|---|---|---|
| | **Model** | | **Cov.** | **MRR** | **Cov.** | **MRR** | **Cov.** | **MRR** | **Cov.** | **MRR** |
| **Baselines** | **Exact Match** | | 25.2% | 0.252 | 25.2% | 0.252 | 25.2% | 0.252 | 25.2% | 0.252 |
| | **Partial Match** | | 88.8% | 0.301 | 78.2% | 0.311 | 51.3% | 0.297 | 21.1% | 0.211 |
| | **Char 4-Gram** | | 93.6% | 0.312 | **81.3%** | 0.313 | 51.7% | 0.301 | 21.8% | 0.218 |
| | **BM25** | | 78.9% | 0.376 | 72.7% | 0.371 | 53.0% | 0.364 | 29.0% | 0.290 |
| **DiscK-1** | **Weighted** | **R = 0.4** | 90.0% | **0.559** | 80.9% | 0.558 | **66.9%** | **0.555** | **50.2%** | **0.502** |
| | | **R = 0.25** | 90.0% | 0.556 | 81.0% | **0.559** | 66.4% | 0.549 | 49.8% | 0.498 |
| | **Binary** | **R = 0.4** | 89.2% | 0.465 | 79.6% | 0.467 | 59.8% | 0.457 | 37.5% | 0.375 |
| | | **R = 0.25** | 89.2% | 0.459 | 78.9% | 0.464 | 61.4% | 0.452 | 39.1% | 0.391 |
| | **Combined** | **R = 0.4** | **95.3%** | 0.357 | 78.3% | 0.362 | 48.0% | 0.342 | 28.6% | 0.287 |
| **DiscK-2** | **Weighted** | **R = 0.4** | 92.8% | 0.476 | 80.3% | 0.476 | 60.0% | 0.468 | 42.0% | 0.420 |
| | | **R = 0.25** | 92.9% | 0.469 | 80.3% | 0.469 | 58.7% | 0.459 | 41.3% | 0.413 |
| | **Binary** | **R = 0.4** | 89.5% | 0.448 | 79.0% | 0.455 | 59.9% | 0.438 | 37.3% | 0.373 |
| | | **R = 0.25** | 89.5% | 0.448 | 79.5% | 0.441 | 60.2% | 0.450 | 36.4% | 0.364 |

Table 1: Coverage (the percentage of instances that the relevant concept was generated) and Mean Reciprocal Rank (MRR) for DiscK and Baselines, for varying candidate list sizes $K$ on the test data. The DiscK models are described in Section 4 and the Baseline models in Section 6. The DiscK-1 is the best performing model in all cases for MRR, and for coverage at $K = 10$ and $K = 1$. For coverage at $K = 1000$, the Combined model performs best, and at $K = 100$, the char 4-gram model is the best.

| Model | p-value |
|---|---|
| DiscK-1, R = 0.25 | 2.20e-8 |
| DiscK-1, R = 0.4 | 1.99e-8 |
| DiscK-2, R = 0.25 | 0.077 |
| DiscK-2, R = 0.4 | 0.002 |

Table 2: Wilcoxon signed-rank test comparing DiscK-1 weighted and binary rankings, using the models shown in Table 1. With the exception of *DiscK-2, R = 0.25*, all have a p-value $< 0.01$, which shows a significant difference between the weighted and binary rankings of DiscK.

systems assign the relevant concept a high score, they also produce candidate lists with many ties, which reduces the usefulness of the ranking. The effect of this can also been seen in coverage with smaller candidate list sizes (e.g. 10). The discriminative ranking of the DiscK models results in relevant concepts receiving higher relative scores, resulting in better coverage for smaller list sizes.

The Combined model explores whether the benefits of the DiscK-1 model (high mean reciprocal rank) can be combined with the benefits of the Character 4-gram model (high coverage). This model provides a mean reciprocal rank that is higher than that in the Character 4-gram model, and the highest level of coverage for $K = 1000$ of any model. However, the addition of the Character 4-gram candidates increases the amount of noise in the candidate list, and thus while the coverage is competitive at higher levels, the MRR is consistently lower than the DiscK models alone.

| Model | Coverage | MRR |
|---|---|---|
| Full model | 92.9% | 0.469 |
| − Ontology aug. | 84.3% | 0.394 |
| − Exact Match | 92.1% | 0.324 |
| − Partial Match | 92.5% | 0.470 |
| − Lemma | 92.1% | 0.468 |
| − Partial & Lemma | 91.7% | 0.441 |
| − Char 4-Gram | 89.9% | 0.497 |
| − Partial, Lemma & Char 4-Gram | 43.8% | 0.298 |
| − Definition | 91.5% | 0.432 |
| − Abbreviation | 92.2% | 0.468 |

Table 3: Feature ablation results for one DiscK model (DiscK-2, R = 0.25, N = 1000) on the test set. The change in coverage and MRR is shown for the removal of each feature or set of features. The features are described in Section 4.

The baseline and feature ablation results also show that lexical matching algorithms can provide a high level of coverage in generating candidate lists. With partial matching, for example, 88.8% of relevant concepts are retrieved, which is competitive with the best DiscK model. As seen in Table 3, the removal of partial, lemma, and character-gram matching (as they often provide similar information) reduces the coverage to only 43.8%. However, the performance of lexical matching algorithms is partially deceiving, as many lexical matches are made with the augmented synonyms. Without the augmentation step, the effectiveness of DiscK model 2 drops to 84.3% coverage. Without this step, many concept links would require non-lexical transformation.

In reviewing the coverage errors for the DiscK-2 model (R = 0.25, N = 1000), which provides the highest coverage, several patterns emerged. First, 25 of the 76 errors would require non-lexical transformation to match the mention and concept. An additional 26 could achieve a partial match with some lexical transformation, but some tokens would require non-lexical transformation. The remaining 25 errors could achieve a match with the correct lexical transformation – most were not retrieved due to morphology or abbreviation. Additional non-lexical errors were avoided due to the synonym augmentation step. However, this is less useful when applying this solution to a larger dataset with a bigger vocabulary, as many non-lexical transformations may not have been seen in the training data.

## 8. Concept Linking Improvements

To demonstrate the effectiveness of our candidate generation system, we used it in conjunction with an end-to-end concept linking system. We selected DNorm [Leaman et al., 2013], a concept linking system that builds weighted TF-IDF representations of both the mention string and concepts, and learns a weighted similarity measure to rank concepts. DNorm was the highest performing concept linking system in the Share/CLEF 2013 task [Pradhan et al., 2013]. While DNorm is accurate, it must calculate the similarity between each mention and every concept in the knowledge base. For our dataset, the number of candidates is $n = 125,362$. To reduce the number of concepts

|  |  | DNorm Ranked List Size $d$ |  |  |  |  |  |
|---|---|---|---|---|---|---|---|
|  |  | 1 | 10 | 50 | 100 | 500 | 1000 |
| | 50 | 61.92% | 67.78% | 73.29% | | | |
| | 100 | 64.07% | 72.81% | 77.84% | 78.80% | | |
| | 250 | 69.09% | 78.87% | 84.13% | 85.32% | 86.28% | |
| Candidate List Size $k$ | 500 | 71.03% | 80.21% | 86.17% | 87.49% | 88.65% | |
| | 1000 | 70.60% | 81.31% | 87.86% | 89.17% | 90.24% | 90.83% |
| | 2000 | 71.67% | 82.26% | 87.62% | 90.36% | 91.55% | 92.38% |
| | 5000 | 71.90% | 81.90% | 88.74% | 90.95% | 92.74% | 93.57% |
| | **All** | **74.79%** | **83.23%** | **88.94%** | **92.03%** | **94.88%** | **95.48%** |

Table 4: Coverage results for DNorm using varying Candidate List sizes $k$. Cells for DNorm's ranked list with a size larger than $k$ are left empty.

considered, we used our candidate generation method to filter concepts evaluated by DNorm. As DiscK retrieval is a sublinear operation, generating candidates in this manner is more efficient. We now operate over candidate lists of (at most) size $k$ [7], instead of the full knowledge base, a significant gain in efficiency.

We use the DiscK-2 model (as it had the highest coverage at 1000), [8] and we retrained DNorm using the train, development, and tests splits noted in Section 5. In Table 4, we report the coverage for varying $k$ and for varying sizes of DNorm ranked lists of size $d$. We do not report improvements in terms of time, as DiscK could not be directly integrated into DNorm. Table 5 reports mean reciprocal rank. Since our candidate generation method eliminates some correct concepts from consideration, the DNorm version with filtered concepts performs worse than if considering all candidate sizes. However, the difference in accuracy is small for larger candidate sizes – for $k = 5000$, the accuracy at $d = 1000$ is only 1.97% worse than when considering all candidates. Similarly, the mean reciprocal rank for $d = 1000$ is 0.023 points lower than when consider all candidates. For this small reduction in performance we see dramatic speedups; DNorm with a candidate list of size $k = 5000$ only considers 4% of the original candidates. For smaller levels of $k$, the performance in terms of coverage and mean reciprocal rank continue to decrease, but are paired with larger gains in efficiency. In more extreme cases, such as that of $k = 50$ and $d = 50$, there is a 15.7% decrease in accuracy, but only considers 0.04% of the original candidates. These large gains in efficiency are particularly attractive when performing concept linking over a large corpus, such as the EHR of a large hospital.

## 9. Related Work

Previous work in Medical Concept Linking focused on building end-to-end systems, which combine candidate generation and final linking. Metamap [Aronson, 2001, Aronson and Lang, 2010] consists of a pipeline to detect candidate spans and link concepts to the UMLS. The system consists of a pipeline that also performs pre-processing tasks, such as tokenization, negation detection, word

---

7. The candidate lists generated by DiscK will contain at most $k$ candidates, but may contain less if fewer matches are retrieved

8. The character model requires pairwise comparisons, so it would not improve DNorm efficiency.

|   | | **DNorm Ranked List Size** $d$ | | | | | |
|---|---|---|---|---|---|---|---|
|   |       | 1     | 10    | 50    | 100   | 500   | 1000  |
| **Candidate List Size** $k$ | 50    | 0.619 | 0.643 | 0.643 |       |       |       |
|   | 100   | 0.641 | 0.674 | 0.676 | 0.676 |       |       |
|   | 250   | 0.690 | 0.724 | 0.726 | 0.727 | 0.726 |       |
|   | 500   | 0.710 | 0.740 | 0.740 | 0.744 | 0.743 |       |
|   | 1000  | 0.706 | 0.742 | 0.745 | 0.746 | 0.746 | 0.744 |
|   | 2000  | 0.717 | 0.754 | 0.755 | 0.758 | 0.757 | 0.759 |
|   | 5000  | 0.719 | 0.755 | 0.757 | 0.757 | 0.757 | 0.758 |
|   | **All** | **0.748** | **0.777** | **0.781** | **0.782** | **0.781** | **0.781** |

Table 5: Mean Reciprocal Rank results for DNorm using varying Candidate List sizes $k$. Cells for DNorm's ranked list with a size larger than $k$ are left empty.

sense disambiguation, and named entity recognition, to identify potential candidate mentions. The candidate generation approach used in the original version consists of generating a candidate list consisting of concepts that containing a variation of the mention phrase. These are then scored by an evaluation function which considers the type of variation – spelling variants are not penalized, while derivational variants are the most penalized. The CTakes medical natural language processing pipeline [Kipper-Schuler et al., 2008, Savova et al., 2010] consists of a similar set of natural language processing tools to process clinical notes, and includes concept linking. The original system used a dictionary matching algorithm to match mention spans to entries in the ontologies and their variant forms.

A newer medical concept linking system uses a concept generation and final linker configuration - in Aggarwal and Barker [2015], they generate candidates from concepts containing variants of the tokens in the mention text, weighing them by inverse document frequency. The candidate list is then re-ranked by the similarity between the mention and candidate context, defined as bag of words in the mention sentence and concept definition. Many medical concept linking systems do not include a distinct candidate generation phase. This includes a Sieve-Based method [D'Souza and Ng, 2015] which uses an ordered set of rules to identify a matching concept. This system does not include a separate candidate generation phrase, but relies on a set of high-precision rules to match mentions to concepts from the entire set of candidates from the ontology. Another proposed system [Rajani et al., 2017] combines the output of several systems. They then train a system to learn the strengths and weakness (e.g. that a system is very precise, but has poor recall) of each by using auxiliary features, such as context-concept similarity. Finally, they ensemble the output of all systems by also considering which system is best suited for a specific mention.

Many systems have focused on the related task of Bio-medical literature concept linking [Doğan et al., 2014, Zheng et al., 2015, Tsai and Roth, 2016], using a pairwise ranking approach, abstract meaning representation, and an indirectly supervised ranking approach, respectively. Additional work has focused on non-medical approaches to similar tasks, specifically in Entity Linking. Instead of linking to concepts, mentions containing entities, such as persons or places, are linked to a knowledge base containing standardized forms of those entities [Cucerzan, 2007, Dredze et al., 2010, Shen et al., 2015, Cheng and Roth, 2013, Andrews et al., 2014, Liu et al., 2013, Pan et al., 2015, Tsai and Roth, 2016].

## 10. Conclusion

For medical concept linking, using a weighted feature-based candidate generation step produces a more robust candidate list than standard triage steps. Compared to baselines, we find that DiscK produces a candidate list that has a high level of coverage, but also ranks the relevant concept higher than standard methods. This approach provides improved input for a final linking method, as the DiscK candidate list better disambiguates between relevant concepts and non-relevant concepts. We find that the majority of concept links can be identified with lexical features, but identifying concepts that are not lexically similar requires additional investigation. Integration of our candidate generation step into an existing concept linking program [Leaman et al., 2013] shows that with a small reduction in accuracy, large efficiency gains can be made by replacing a complete pairwise search of the possible candidates with the sublinear DiscK candidate generation system.

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
