# OpenReview forum: "Discriminative Candidate Generation for Medical Concept Linking"
_AKBC.ws/2019/Conference — AKBC 2019_

### Official Review · AnonReviewer1 · 2018-12-16
**throrough study and a good reminder to think about this important step in the entity linking pipeline**

**Rating:** 9
**Confidence:** 5

**Review:**

In the area of entity linking, this paper focuses on candidate generation. The author make a valid point, for the importance of this stage: While much work focuses on disambiguation of a candidate set of entities, this only matters if the correct choice is present in the candidate set. This might not be true for simple candidate generation methods. The paper suggests a weighted combination of different candidate generating features, such as exact string match, BM25, character 4-grams or abbreviation matches. These are contrasted with two variations of the DiscK approach.

The evaluation is based on the Share/Clef eHealth task. It is very exhaustive, evaluating different properties of the candidate set. From an efficiency standpoint, I  would be interested in candidate methods that contain the true candidate in a small pool set i.e., 1000 might not be affordable in a real system. I only have one doubt: The lines do not cross for MRR  (Figure 2), so how is it possible that they cross for  coverage (Figure 1)? Maybe standard error bars would be helpful to understand which differences are due to quantization artifacts.

It would have been useful to study the effects of the candidate set on the final result when combined with a candidate-based entity linker.

I am not sure I follow the terminology of (DiscK), weighted, combined, binary -- it would help to have an exhaustive list of methods with consistent names.

If the authors find it useful, here is a related paper that also discusses the effects of the candidate method during entity linking: Dalton, Jeffrey, and Laura Dietz. "A neighborhood relevance model for entity linking." Proceedings of the 10th Conference on Open Research Areas in Information Retrieval. 2013.

---

> ### Author Response · Authors · 2019-01-23
> **Response to Reviewer 3**
>
> Thanks for your comments.  As you point out, the lines do not cross for MRR but do cross for coverage.   This is due to the fact that while the Char 4-gram has higher coverage at high K levels, the correct candidates are still ranked very low, and have a low effect on the MRR.  DiscK overall does a much better job of ranking correct concepts highly, but omits some that Char 4-gram assigns a low ranking.
>
> We agree that the size and contents of the candidate set would have significant implications for the final linking results.  We are limited in the flexibility we have to experiment with this, since the candidate set is determined by the annotators, but it is an important point.  Regarding the terminology, DiscK weighted uses the parameters learned by the training procedure (discussed in Response #1), while DiscK binary only considers the presence of features (i.e. all parameters are 1) and does not use the learned weights.  The combined model uses the best DiscK weighted with a model that has higher coverage at K=1000 (Char 4-gram).  For each candidate, we take the higher of the two normalized scores to achieve a combined ranking.

---

### Official Review · AnonReviewer3 · 2018-12-23
**Some efficiency gains, but more rigorous evaluation and motivation required**

**Rating:** 5
**Confidence:** 3

**Review:**

This paper proposes using an existing method, DiscK, to pre-filter candidate concepts for a medical concept linking system. When combined with an existing state-of-the-art linker (DNorm), the joint system is able to consider much fewer candidate concepts, albeit with some drop in accuracy. These results are potentially significant, but assessing the actual impact requires additional measurements and better explanation of the motivation.

Pros:
- DiscK + DNorm achieves concept linking accuracy close to the previous state-of-the-art, while requiring DNorm to process significantly fewer candidates. For example, the system gets up to .759 MRR when considering 2000 candidates, compared to .781 MRR for the full set of ~125K candidates (Table 5).

Cons:
- Evaluation of candidate generation without downstream linking (Section 6) was not compelling. Without information about how the linker might use these candidate sets, it is hard to tell whether these numbers are good or not.
- The claims of superior computational efficiency in Section 8 are not supported quantitatively. There should be a comparison between DNorm alone and DiscK + DNorm of how many mentions or documents can be processed per second. Otherwise, it is not clear how much overhead DiscK adds compared to the cost of running DNorm on more candidates. Hopefully this issue can be easily addressed.
- Writing was at times unclear and misleading. Most importantly, the authors write, "DiscK allows us to efficiently retrieve candidates over a large ontology in linear time." In fact the point of DiscK is that it enables sublinear queries in the size of the candidate space/ontology. But perhaps the authors are referring to being linear in something else? After all, pairwise scoring models are also linear in the size of the candidate space.
- Straightforward application of existing approaches, with no new technical contribution

Additional comments:
- As an outsider to this medical linking task, I wanted to understand more why speed is important. How expensive is it to run DNorm on a large corpus? Are there online settings in which low latency is desired, or is this only for offline processing?
- I would like some discussion of two other seemingly more obvious ways to improve runtime:
    (1) Learn a fast pairwise comparison model and filter using that. The number of candidates is large but not unmanageably large (125K), so it seems possible that you could still get significant wall clock speedups by doing this (especially as this is very parallelizable). Such an approach would also not have to abide by the restrictions imposed by the DiscK framework.
    (2) Speed up DNorm directly. Based on the original paper (Leaman et al., 2013), it seems that DNorm is slow primarily because of the use of high-dimensional tf-idf vectors. Is this correct? If so, might a simple dimensionality reduction technique, or use of lower-dimensional word vectors (e.g. from word2vec) already make DNorm much more efficient, and thus obviate the need for incorporating DiscK? Relatedly, are other published linkers as slow as DNorm, or much faster?
- Based on table 5, in practice you would probably choose a relatively high value of K (say ~2000), to maintain near state-of-the-art accuracy. We also know that at high values of K, character 4-gram is competitive with or even better than DiscK. So, what is the runtime profile of the character 4-gram approach? Footnote 8 mentions that it is asymptotically slower, but computing these character overlaps should be very fast, so perhaps speed is not actually a big issue (especially compared to the time it takes DNorm to process the remaining K candidates). This is is related to point (1) above.
- The authors mention that DNorm is state-of-the-art, but they don't provide context for how well other approaches do on this linking task. It would be good to know whether combining DiscK + DNorm is competitive with say, the 2nd or 3rd-best approaches.
- The organization of this paper was strange. In particular, Section 8 had the most important results, but was put after Discussion (Section 7) and was not integrated into the actual "Evaluation and Results" section (Section 6).
- Measuring MRR in Section 6 is unmotivated if you claim you're only evaluating candidate generation, and plan to re-rank all the candidate sets with another system anyways. It is still good to report these MRR numbers, so that they can be compared with the numbers in Section 8, so perhaps all of the MRR's should be reported together.

---

> ### Author Response · Authors · 2019-01-23
> **Response to Reviewer 2**
>
> Thanks for the comments.  While we do not report downstream results in Section 6, we do in Section 8.  We did so in this setup so that we could first compare potential retrieval methods, and then test the DiscK method downstream.  We agree that reporting timed results would be ideal.  However, after a significant amount of engineering effort, we were unable to directly integrate the DiscK method into DNorm, as it is an older codebase.  Therefore, we run the DiscK retrieval method, save the results, and finally loading the results into DNorm.  This doesn’t allow for a fair comparison on timed efficiency, so we are left with the efficiency measurements we report in the paper.  In addition, thanks for correctly pointing out that instead of linear time, we should state that DiscK is sublinear time - this will be corrected in the paper.
>
> In response to comment (1), it is true that the size of this dataset (125k concepts) could possibly allow for a pairwise model, as you suggest.  However, the mentions in this dataset are only linked to a subset of one ontology, SNOMED Disorders.  In real clinical settings, the size of the ontology is often much larger, as the entire ontology (SNOMED has 300K concepts) or several ontologies may be used in combination, leading to a concept set of 1 million or more.  Therefore, we wanted to approach this problem in a way that will scale to very large ontologies, and a pairwise approach, while potentially feasible for this dataset, would get computationally expensive quickly for larger ontology sets.  The downside of the pairwise approach extends to DNorm, which essentially performs a weighted similarity calculation between the tf-idf vector of the mention and the concept.  In larger settings, this will grow very slowly.  While most of the time concept linking is performed offline, there are several scenarios where online concept linking would be useful, such as a clinician seeking to process a specific clinical note.
>
> Following off of the above point, this is part of the reason we don’t simply make DNorm more efficient (as suggested in (2)), as it is still a pairwise method.  Furthermore, we wanted to propose a general method that would allow candidate generation for any system, and not simply speed up an existing one.
>
> As noted in the response to reviewer #2, the DNorm system was one of the best performing systems on the task (at the release of the shared task results, see Pradhan et al 2015), so that point of comparison is available.  However, those results are on the test set, and aren’t directly comparable.  While we don’t currently have the code for any of the other systems reported in that paper, we do have access to a later system (“Sieve-Based Entity Linking for the Biomedical Domain”, D’Souza and Ng 2015), and we can use that as a point of comparison for the final paper.

---

### Official Review · AnonReviewer2 · 2019-01-08
**Compelling initial result but would benefit from additional experiments, especially direct comparisons with prior work.**

**Rating:** 5
**Confidence:** 4

**Review:**

This work applies an existing system, DiscK (Chen and Van Durme, 2017) to medical concept linking. They show that DiscK improves the mean reciprocal rank of candidate generation over several simple baselines, and leads to slightly worse but significantly more efficient linking performance in one existing downstream system, DNorm (Leaman et al., 2013).

Pro’s:
-	Clear description of the task and associated challenges.
-	Thorough explanation of the proposed system, experiments, and results. In particular, the discussion of the performance trade-off between MRR and coverage, and why it is useful that DiscK maintains robust MRR even though it under-performs several baselines w.r.t. coverage, was clear and useful.
-	The downstream concept linking result is compelling: for example, the DNorm+DiscK system with k=5000 and d=1000 only considers 4% of candidates but its coverage declines by just 2% compared to considering all 100% of candidates (as in DNorm alone).

Con’s/Suggestions:
-	Section 3/DiscK Overview: Without having read the DiscK paper in full (admittedly), I found Section 3 very hard to follow. How are the feature weights chosen? How are the feature parameters trained? What’s the difference between a feature type and a feature value? Since the DiscK work is integral to the rest of the paper, the authors should spend more time giving a high-level overview of the approach (targeted at “naïve readers”) before delving into the details.

-	Section 4: The authors note at the end of Section 4 that “While we found that many mentions could be matched to concepts by lexical features, a significant portion required non-lexical features.” It would be helpful if the authors provided a concrete example of such a case.

Also, they note that “While features from additional mention properties, such as the surrounding sentence, were tested, none provided an improvement over features build from the mention span text alone.” This merits more detailed explication: What features did they try? Since examples requiring non-lexical features were a consistent source of error, why do the authors think that non-span-based features failed to influence model performance on these examples?

-	Data: It is unclear to me whether expanding the ontology to include the additional concepts (e.g. Finding) would prevent comparison with other systems for this dataset, and what is gained from this decision. In addition, it would be helpful if the authors explained what a “preferred entry” is in the context of the SNOWMED-CT ontology.

-	Evaluation & Results:

The authors note that “The concept linking systems that were evaluated in the shared task may have included a candidate generation stage, but evaluations of these stages are not provided.” However, it is unclear to me why the systems that do include a candidate generation phase could not either be re-run or re-implemented to get such results, especially since Share/CLEF is a shared task with many entries and corresponding published system descriptions. Since direct comparisons with prior work were omitted, it is hard to gauge the strength of the baselines and proposed system.

In addition, it might be useful to test the value of higher MRR in the downstream concept linking task by comparing the DiscK-based candidate generator against one of the baseline approaches with higher coverage but lower MRR. Can DNorm capitalize on a more informative ranking, or are the results similar as long as a basic level of coverage is achieved for a given k?

-	Related Work: The authors only briefly mention entity linking work outside of the biomedical domain. Did the authors evaluate any of these approaches, especially in the context of computationally-efficient linking architectures? Also, a more detailed description of DNorm would be useful.

---

> ### Author Response · Authors · 2019-01-23
> **Response to reviewer 1**
>
> Thanks for your comments on our work.  First, we can clarify several points on DiscK.  The goal of DiscK is to use a simple feature set to learn a weighted similarity between a query and a set of documents.  In our task, the query is a mention and the set of documents is a set of concepts from the UMLS.  It is parameterized in a way that allows for retrieval in sublinear time (as reviewer #2 points out, we incorrectly said “linear” - this will be corrected to “sublinear”).
>
> For the training procedure, the weights θ are trained using a negative sampling procedure, with the goal of learning a set of weights that will correctly predict mention-concept pairs.  For each training mention, it is paired with the correct ontology entry and 50 incorrect ontology entries.  The resulting weights are used to project which ontology entry is most suited to the mention feature set.  This is trained using a log-linear model.
>
> The feature set we explored that included additional mention properties (e.g. the surrounding sentence and the definition of the concept, if available) used the same approach as the mention features, except that they included the text of the sentence or definition.  For example, one proposed feature was a bag-of-words approach using the surrounding sentence and the concept definition.
>
> Regarding the data, we cannot directly compare to previously reported results on this dataset for several reasons.  As pointed out, we do use a specific subset of the ontology, and it isn’t clear which specific subsets other papers used to evaluate. However, the more important point is that we do not have access to the test set (as noted in Section 5), and therefore we don’t have the necessary data for a direct comparison.  The DNorm system was one of the best performing systems on the task (at the release of the shared task results, see Pradhan et al 2015), so that point of comparison is available.

---

### Meta-Review · Area_Chair1 · 2019-02-11
**Addresses an important, but under-investigated subproblem of entity linking**

**Recommendation:** Accept (Poster)
**Confidence:** 4

**Metareview:**

Summary: while the reviewer opinion on this paper varied,  the paper tackles an important problem while also bringing to light an existing technique that many have overlooked for this problem.

The main strength of this paper is that it addresses an oft overlooked sub-problem of entity-linking: the candidate generation stage prior to linking (though note, that it is not uncommon for entity linking papers to also evaluate their candidate generation system separately from their linker).  They show that when integrating their candidate generation step into an existing entity linker, they can achieve similar accuracy while gaining efficiency by the more aggressive candidate pruning.  The method is based on DiscK which is in turn based on query expansion approaches from IR that supports discriminative training, but with more efficient search algorithms that run in sublinear time (at the cost of a limited set of available feature types).  The paper shows that the method works better than classic IR baselines such as BM25. There are enough people in our community who would be unfamiliar with these sublinear IR-based classification strategies that the paper could be useful for anyone building a classic entity linking system.

The main criticisms of this paper are that it lacks clarity and novelty; the description of the DiscK algorithm is especially unclear, and ultimately, their approach a straightforward application of DiscK to candidate generation.  There were a few criticisms about the experiments but the authors seem to have addressed these in the response.

---

### Decision · Program_Chairs · 2019-02-15
**AKBC 2019 Conference Decision**

Accept